# Front-of-Package Labels on Unhealthy Packaged Foods in India: Evidence from a Randomized Field Experiment

**DOI:** 10.3390/nu14153128

**Published:** 2022-07-29

**Authors:** S. K. Singh, Lindsey Smith Taillie, Ashish Gupta, Maxime Bercholz, Barry Popkin, Nandita Murukutla

**Affiliations:** 1Department of Survey Research and Data Analytics, International Institute for Population Sciences, Deemed University, Mumbai 400088, India; 2Department of Nutrition, Gillings School of Global Public Health, Carolina Population Center, University of North Carolina at Chapel Hill, Chapel Hill, NC 27516, USA; popkin@unc.edu; 3Vital Strategies, New York, NY 27599, USA; agupta@vitalstrategies.org (A.G.); nmurukutla@vitalstrategies.org (N.M.); 4Carolina Population Center, University of North Carolina at Chapel Hill, Chapel Hill, NC 27516, USA; bercholz@email.unc.edu

**Keywords:** warning labels, Health Star Rating, Nutriscore, GDA, food policy, obesity prevention, non-communicable diseases

## Abstract

Policies to require front-of-package labels (FOPLs) on packaged foods may help Indian consumers to better identify foods high in nutrients of concern, including sugar, saturated fat, and sodium, and discourage their consumption, which are outcomes that are critical for preventing rises in diet-related non-communicable disease. The objective was to test whether FOPLs helped Indian consumers identify “high-in” packaged foods and reduce intentions to purchase them. We conducted an in-person randomized experiment (*n* = 2869 adults between ages 18 and 60 years old) in six states of India in 2022. Participants were randomized to one of five FOPLs: a control label (barcode), warning label (octagon with “High in [nutrient]”), Health Star Rating (HSR), Guideline Daily Amount (GDA), or traffic light label. Participants then viewed a series of packaged foods high in sugar, saturated fat, or sodium with the assigned FOPL, and rated product perceptions and label reactions. Fewer than half of participants in the control group (39.1%) correctly identified all products high in nutrient(s) of concern. All FOPLs led to an increase in this outcome, with the biggest differences observed for the warning label (60.8%, *p* < 0.001), followed by the traffic light label (54.8%, *p* < 0.001), GDA (55.0%, *p* < 0.001), and HSR (45.0%, *p* < 0.01). While no FOPLs led to a reduction in intentions to purchase the packaged foods, the overall pattern of results suggested that warning labels are the most effective FOPL to help Indian consumers identify unhealthy foods.

## 1. Introduction

Over the past several decades, with the emergence of the epidemiological transition, India has experienced a growing problem of overweight and obesity and all the major nutrition-related noncommunicable diseases, especially diabetes and hypertension [1,2,3]. According to the latest National Family Health survey, nearly 1 in 4 adults and 1 in 20 children are classified as overweight or obese [4]. Rates are increasing faster in India than the world average, and obesity prevalence is expected to more than triple by the year 2040 without intervention [5,6,7]. At the same time, India faces a major double burden of malnutrition, as stunting and other forms of undernutrition remain high among the rural poor, in particular [4,8,9,10].

These changes have occurred at a time when a remarkable diet transformation is occurring in India, which affects rich and poor, young and old. In particular, the growth of ultra-processed food consumption in India is significant. As shown in neighboring Nepal, even preschoolers are increasingly being fed these foods, as one study found that 25% of preschoolers’ caloric intake came from ultra-processed food and this was linked with higher levels of stunting [11]. From 2006 to 2019, sales of ultra-processed snack food and sweetened beverages in India grew from 1 billion USD to 38 billion USD [12].

Many ready-to-eat or ready-to-heat foods and drinks are high in added sugars, sodium, saturated fats, and refined carbohydrates. Excessive consumption of these nutrients and ultra-processed foods increases the risk of obesity and related NCDs [13,14,15,16,17,18,19,20,21,22,23,24,25]. Indeed, a growing literature including both a large randomized controlled trial and over 45 longitudinal cohort studies linked ultra-processed food with increased risk of overweight/obesity, diet-related NCDs, and total and heart disease-linked mortality [16,26].

To reduce the consumption of packaged foods high in added sugar, sodium, saturated fat, and trans fats, front-of-package labels (FOPLs) have been recommended by the World Health Organization (WHO), the World Bank, and others [24,25,27,28,29,30,31,32,33]. The primary goals of front-of-package labels (FOPLs) are to inform consumers about the nutritional quality of food in a way that is quick and easy to understand and improve the nutritional quality of food purchases, with a secondary goal of stimulating reformulation in the food supply [34]. Interpretive FOPLs are particularly promising because they not only provide information about nutritional content but also help consumers to judge the healthfulness (or unhealthfulness) of products and provide guidance (encouragement or discouragement) regarding the decision to purchase. These FOPLs are important to reduce the intake of the major unhealthy processed foods consumed in India. By synthesizing complex nutrition facts into interpretable information, these labels may be especially valuable for populations with low literacy.

The evidence base on FOPLs is growing rapidly. Warning labels perhaps have the strongest evidence with regard to discouraging purchases of foods high in nutrients of concern, with recent systematic reviews of experimental and quasi-experimental data showing that warnings reduce the selection of unhealthy products by 26% to 36% [35,36]. Another recent review, which was focused on sugar, found that warnings were the most effective at increasing consumers’ understanding of the high nutrient content in foods [37]. Real-world evidence from Chile, the first country to implement mandatory front-of-pack warnings, found that warning labels were linked to a 24% decrease in purchases of unhealthy foods [38] and helped both parents and children identify unhealthy food and drinks and discourage their consumption [39].

In contrast, there is limited real-world data about the effectiveness of other common interpretive FOPLs, such as traffic light labels or the Health Star Rating (HSR) system. Real-world data on traffic light labeling systems has been mixed: one UK-based study found a sizeable reduction in calories purchased linked to the traffic light policy [40], while another study found no association with purchases [41]. Data from Ecuador, which implemented a mandatory traffic light labeling system in 2014, found low self-reported use of the labeling system [42] and no evidence that traffic light labels influenced purchasing behaviors [43,44]. Real-world evidence of the effectiveness of the HSR system has also been scarce. Data from Australia, which implemented voluntary HSR schemes in 2014, show low uptake of the HSR, with implementation skewed toward products considered to be healthier (i.e., with higher ratings) [45]. To our knowledge, there is no evidence from experimental studies or real-world evaluation studies that HSR leads to healthier food purchases [46,47,48]. Evidence in favor of the industry-promoted Guideline Daily Amount (GDA) system is the weakest of all, with an array of both experimental and real-world evaluation studies from across the globe finding that, relative to almost all other FOPL types, the GDAs are poorly understood, take the most time to evaluate, and are the least effective at influencing purchases [37,47,49,50,51,52,53,54,55,56,57,58]. 

However, at the time this study was planned, there was virtually no evidence about what FOPL system will work best to inform Indian consumers about packaged foods that are excessive in nutrients of concern and discourage purchases of these products. A literature review of recent studies found narrative reviews [59] and commentaries [60] supporting the introduction of FOPLs as a policy to curb ultra-processed food intake in India. The only empirical study was an online report that found that, in a randomized experiment of 20,564 Indian consumers, warning labels appeared to perform best at reducing purchasing intentions toward unhealthy packaged foods [61]. The study found mixed results on an array of secondary outcomes, but the pattern of results suggested that warning labels and the HSR label performed better than other FOPL types (traffic lights, GDA) with regard to participants’ ranking of labels on various dimensions, such as ease of identification and ease of understanding. However, the lack of transparency about the methodology used and selection of outcomes warrants additional empirical research to understand which FOPLs work best in the Indian population. 

In this context, the objective of this study was to experimentally evaluate the impact of FOPLs on consumers’ ability to correctly identify packaged products as containing excess levels of nutrients of concern and intentions to purchase them relative to a control label in a sample of Indian adults across six states. Secondary outcomes included consumers’ reactions to the FOPLs and perceptions of unhealthy packaged products.

Given the diversity of the Indian population in terms of language, culture, dietary intake, and educational attainment, it is also essential to ensure that any FOPL regulation works well across the entire population, as well as for different food categories. To address this, we explored whether the impact of FOPLs varied by product type, educational level, and state.

## 2. Materials and Methods

### 2.1. Institutional Review Board Approvals

This study was reviewed and approved by the Institutional Review Board (IRB) at the International Institute for Population Sciences (IIPS) in Mumbai, India, and by the Biomedical Research Alliance of New York (BRANY), which is a national organization that provides IRB services. 

This study was pre-registered at the Open Science Framework on 23 December 2021: https://osf.io/8kx3e. De-identified data is available here: https://osf.io/r3h9w/. Participants provided written consent or, for those who could not provide a signature, verbal consent.

### 2.2. Setting 

We carried out an in-person field experiment in rural and urban areas of 6 states (Assam, Delhi, Gujarat, Odisha, Karnataka, Uttar Pradesh) from January to March of 2022. These states were chosen purposively as sentinel sites to represent the geographic areas of India, as well as key associated sociodemographic variations. From each of these states, one district was selected (Delhi, Mysuru, Bhubaneshwar, Lucknow, Ahmedabad, Guwahati).

First, four wards (two urban, one semi-urban, one peri-urban) were randomly chosen from each district. Next, a listing of potential survey locations was undertaken in each of these selected wards. Each location was classified into four clusters, namely, peak-day peak-time, peak-day lean-time, lean-day lean-time, and lean-day peak-time. This generated a sampling frame of time-location clusters (TLCs). Four TLCs per ward were randomly selected from the list for the survey. The locations were the places that sold packaged food items. These could either be a shop/retail outlet (called Kirana shops locally), large grocery store (in a shopping mall or on a high street), group of small shops, or smaller petty shops (tea stalls, shops selling *paan*—betel leaves wrapped around tobacco, fruits, etc.).

### 2.3. Participants

The participants were 2869 adults between the ages of 18 and 60 years. Recruitment of participants was done via customer interception. Once the nth person was intercepted, they were checked for eligibility and a request was made for an interview. Interviews were conducted with those who consented. After the interview, the next nth person was intercepted, whereas, after a refusal, the immediate next person was intercepted. Each person who agreed to the interview was taken to a close place for the interview where disturbances from the street were minimal.

Within each state, quota were used to obtain approximately 50% of participants who were women and with an educational level of 12 years or less. Eligibility criteria included being between the ages of 18 and 60 years and being involved in decision making related to grocery purchases for their household at least half the time.

### 2.4. Stimuli

Four FOPLs were selected for testing based on conversations with Indian health advocacy organizations and governmental organizations indicating that these labels were of interest for informing an impending FOPL regulation. In addition, the GDA was selected because it has already been voluntarily implemented on some products in India. Images of the FOPLs (as mocked up on sweet biscuits) are depicted in Figure 1.

Warning label: The main design was modeled on the proposed warning label used in South Africa [62]. A design agency adapted the warning for India through design testing with 15 adults in five cities of India to ensure that the label was noticeable and understandable in a socioeconomically diverse population. The warning label was comprised of a white holding strap with the marker word ALERT! and at least one triangle-shaped warning and up to three warnings, depending on the nutrient content of the product (with text: HIGH IN SUGAR, HIGH IN SODIUM, or HIGH IN SATURATED FAT). Based on prior evidence that icons increase perceived effectiveness and comprehension of the label across populations speaking different languages [63] and literacy levels [62], icons depicting sugar, salt, and saturated fat were also used.

HSR: The HSR was modeled after the existing HSR system used in Australia and New Zealand. The circular label stated HEALTH STAR RATING and depicted a number of stars from 0.5 to 5 shaded in black to indicate the healthfulness of the product, with fewer stars indicating less healthy and more stars indicating healthier.

Traffic light labels: The traffic light label was based on a simplified version of the system used in the UK and Ecuador. The label presented color-coded information on sugar, sodium, and saturated fat for each product, with red signaling high, amber signaling medium, and green signaling low content of that nutrient.

GDA: The GDA was based on the existing GDA used voluntarily by the food industry in many countries. The GDA contained four blue shaded shapes containing nutritional information on calories, saturated fat, sugar, and salt (both the absolute content in calories or grams, as well as the percent of an adult’s guideline daily amount).

Control label: Similar to previous FOPL experimental studies [64,65], a barcode label was used as a control label because it serves as a piece of visual information on the front of the food package while conveying neutral information about the product’s nutritional content.

The labels were displayed on a series of products, including a savory biscuit, a loaf of bread, a fruit drink, a sweet biscuit, and a package of instant noodles (Appendix A
Figure A1). These product categories were chosen because they are commonly consumed in India, are often high in nutrients of concern, and represent categories where there may be high levels of consumer confusion about the nutritional content of the products. A professional designer designed mock products to avoid the influence of brand preferences, though to increase realism, the mock products and their nutritional information were based on popular Indian brands.

For each product, one commercial brand within each food category was selected. A mock nutrient profile was created based on this brand (±2% of the original nutrient profile model). Each label was then based on the relevant nutrient profile model: for HSR, Australia’s HSR calculator was used [66]; for warning labels and traffic light labels, the thresholds specified in the 2019 draft regulation for Food Safety Standards and Authority (FSSAI) were used; and for the GDA, national dietary guidelines were used [67].

### 2.5. Cognitive Testing and Protocol Development

The study protocol and measures used were developed and refined through an iterative process to ensure acceptability among diverse participants. First, the study items were translated from English into five languages (Assamese, Gujarati, Hindi, Kannada, and Odia). Two rounds of cognitive interviews were completed to make sure the measures were properly adapted to the Indian context and well-understood in each language while maintaining consistency with the underlying construct [68]. The interviews were completed in two phases, with each phase including four participants in each language (40 interviews total), with a refinement of the study measures occurring between phases one and two. After the cognitive interviews were complete, the items were refined and new additions were translated and back-translated to English before being reviewed by study co-authors. The field methodology (including recruitment and study implementation) was then pilot tested in a sample of 20 adults in an urban area of Delhi State in December of 2021 before further finalization of the study protocol.

### 2.6. Procedure

Participants were randomized to one of 5 arms: control label, HSR, warning, GDA, or traffic light label using an allocation ratio of 1:1:1:1:1. Participants then viewed a series of images of products in random order with a FOPL on the product according to the assigned arm. In the control condition, all products had the barcode label. In the HSR condition, all products displayed stars. In the warning label condition, products displayed the relevant warning(s) for sugar, sodium, and/or saturated fat. In the GDA condition, all products had a GDA with the relevant nutritional information. In the traffic light condition, products displayed multiple traffic lights with the relevant color code (green, yellow, or red) for each nutrient.

Interviewers showed participants images of products using an A5 size booklet in random order and asked them to assess the product and their reactions to the label. The participant viewed images of all 5 FOPLs and answered questions about which label they preferred. All data were entered into a smartphone app during the interview.

At the end of the study, participants provided demographic information.

### 2.7. Measures

Socio-demographic and behavioral covariates were specified as follows: gender (man/woman), age (18–30 years, 31–40 years, 41 years and older), education (≤12 years of education, >12 years), urbanicity (defined as peri-rural, semi-urban, and urban), and state (Assam, Delhi, Gujarat, Odisha, Karnataka, Uttar Pradesh). Languages included Assamese (Assam), Gujarati (Gujarat), Hindi (Delhi, Uttar Pradesh), Kannada (Karnataka), and Odia (Odisha). Participants were also able to conduct the survey bilingually (in the language of the state and English) if they preferred. The participant’s financial situation was defined as a four-level variable: (1) can pay the bills and buy necessary and additional things, (2) can pay the bills and buy necessary things only, (3) can pay the bills but not buy necessary things, and (4) cannot pay bills. Household income was categorized as <INR 100,000; INR 10,001–25,000; INR 250,001–50,000, and over INR 50,000. The consumption of “high-in” food (sweet biscuits, salty biscuits, bread, and instant noodles) was categorized as never or less than one time per week, 1 time per week, or more than 2 times per week. 

The codebook including product assessment items and label assessment items is available in Appendix A
Table A1).

For all five products, participants rated their perceptions of the product. First, to assess their ability to correctly identify that the product had high contents of nutrients of concern, they answered the question, “Do you think this product has high [nutrient of concern]?” (yes/no). For two products, the sweet biscuit and the instant noodles, the respondents were asked this question twice, one for each nutrient of concern. 

Next, they were asked, “Is this product unhealthy?” (yes/no). If they answered yes, they were asked, “how unhealthy is it?”, with response options ranging from 1 to 3 (very much, somewhat, very little). They were then asked about visual attractiveness (“Do you think this product is visually attractive?”) and intentions to purchase (“Will you purchase this product next week, if it were available?”) with a yes/no response. Those who answered “yes” were asked the follow-up question (“How visually attractive,” or “How likely,” respectively), with two options again ranging from (1) very much to (3) very little.

Participants also completed a label assessment for three of five products (randomly selected). Participants answered whether the label grabbed their attention, made them feel concerned about the health problems of consuming the product, was understandable, taught them anything, was truthful, and was likable. For perceived message effectiveness, participants were asked whether the label made them concerned about the health consequences of consuming the product, made the product seem unpleasant, and made them feel discouraged from wanting to consume the product. For all items, response options were yes/no. If the respondent answered yes, they were then asked “how much…”, with responses ranging from (1) very much to (3) very little.

Finally, participants were asked to compare their labels and select which label (a) most discouraged them from consuming the product, (b) most discouraged them from feeding the product to a child aged 1–12 years, (c) best informed them that the product has high [nutrient], and (d) was the easiest to understand.

### 2.8. Statistical Analysis

First, we recombined some measures. For all items measured on the Likert scale, we combined the agreement item (yes/no) with the strength of agreement item for each person to create a 4-point Likert scale, subsequently recoded from 1 (not at all) to 4 (very much) for a more intuitive interpretation. For perceived message effectiveness, Cronbach’s alpha for the three items was >0.7; therefore, we created a scale that was the average of the three items for each product type. For the primary outcome, the ability to correctly identify products high in nutrients of concern, we specified this a priori as correctly identifying *all* nutrients since two products had multiple nutrients.

We descriptively reported sociodemographic characteristics and examined whether participant demographics differed by study arm using the chi-square test. Then, we descriptively reported unadjusted percentages and means (and standard deviations) for the two primary outcomes by product type.

For all main analyses, we examined differences between the control label and each of the other FOPL conditions. For all outcomes with multiple measurements for each person, we used mixed-effects logistic regression for the correct identification of all “high-in” nutrients and mixed-effects linear regression for all other outcomes, with respondent-level random intercepts to account for repeated measures. Standard errors were clustered by interviewer. While this is not standard procedure (and was not pre-registered as part of our original study plan), this was done as a conservative measure against any violation of the assumption of independence within interviewer, as some interviewers had very high or very low outcome means in their groups of participants for certain outcomes. We included indicator variables for label type (between-subjects) and product type (within-subjects), as well as an interaction of the label type and product type, if significant at the 5% level (as stated in the pre-registered analysis plan). The Holm procedure was used to adjust the *p*-values for multiple comparisons within each outcome (four tests for the four label types compared to the neutral label). This was done so the familywise error rate across the four tests within each outcome would not exceed 0.05, which was the nominal significance level.

To evaluate the most discouraging label, the most informative label, and the label that was easiest to understand, we descriptively reported the percentage of participants that selected each label type as the most discouraging from consuming the product, most discouraging from feeding the product to a child, most informative, and the easiest to understand.

To assess whether the effect of FOPLs on the primary outcomes differed by socio-demographic and behavioral covariates, we conducted exploratory moderation analyses by adding, in turn, each moderator of interest and its interaction with label type to the main model. For potential moderators, we included education and state. We explored moderation by state instead of language (as stated in the pre-registration) since most states used their own language and state-level differences were of interest conceptually due to regional variation in the food supply and dietary behaviors. We also included urbanicity, gender, whether the survey was conducted bilingually, and weekly consumption of the five product types presented to the participants as additional exploratory moderators. As Cronbach’s alpha was < 0.7 for the five consumption measures, separate models were estimated for each product using the corresponding consumption measure as the exploratory moderator. Since separate models were fit for each product, there were no repeated measures and linear and logistic regressions were used. We tested for overall differences in the effect of each label (relative to the control, i.e., the difference in means between a given FOPL group and the control arm) across the levels of each moderator. In line with the main analyses, the standard errors were clustered by interviewer. Unadjusted *p*-Values were used for these exploratory analyses.

Finally, we conducted two sets of sensitivity analyses on the primary outcomes. Because clustering the standard errors by interviewer is not a standard procedure, we present results where we did not cluster the standard errors. Secondly, as an alternative to clustering the standard errors by interviewer, for each primary outcome, we excluded participants who had been interviewed by one of the six interviewers with the highest or lowest three outcome means in their groups of participants.

We used a two-sided significance level of 0.05 to conduct all statistical tests using Stata version 16.1.

## 3. Results

### 3.1. Descriptive Results

The socio-demographic characteristics of the sample are reported in Table 1. No covariates were found to be unbalanced between the study arms. The sample was roughly distributed proportionately across all six states, with about half of the sample in urban areas and a quarter each in semi-urban and peri-rural areas. The sample was comprised of approximately half women and half with an education <12 years, and the majority of the sample were able to pay the bills and buy what they need. Approximately 40% completed the interview in a mixed language.

Overall descriptive results for all label and product assessment outcomes can be found in Appendix A
Table A2. Descriptive results for label choice can be found in Appendix A
Table A3.

Descriptive results for the primary outcomes by product type can be found in Appendix A
Table A4. For the percent of participants who correctly identified all products high in nutrient(s) of concern, there were some observable differences by product type, with the highest percentage of participants in the control group correctly identifying the fruit drink (60.5%) and the lowest percentage identifying the sweet biscuit (24.7%). Thus, the difference between each FOPL and the control arm was smaller for all FOPL types for fruit drinks relative to most other product types. In contrast, the difference between each FOPL and the control arm tended to be higher for noodles and savory biscuits relative to the other products, though this varied somewhat by FOPL type. There were minimal differences by product type regarding intentions to purchase.

Descriptive results for the primary outcomes by state can be found in Appendix A
Table A5. In the control group, Assam and Delhi were the states that had the lowest percentage of participants that correctly identified products as containing high levels of nutrient(s) of concern (25.9% and 24.4%, respectively), while Uttar Pradesh and Karnataka were the highest (54.0% and 56.7%, respectively). For all FOPL types, the difference between each FOPL and the control arm was the smallest for the state of Odisha. For most FOPLs, the difference between each FOPL and the control arm was the greatest for the state of Uttar Pradesh, followed by Gujarat.

There were also observable differences in intentions to purchase products high in nutrients of concern by state. In the control group, intentions to purchase were highest in Assam (3.1, SD 0.8) and lowest in Delhi (2.3, SD 1.1), Gujarat (2.4, SD 1.0), and Odisha (2.4, SD 1.1).

### 3.2. Main Results

Results of the primary outcomes can be found in Figure 2 and Figure 3. Relative to the control group (39.1% of participants; 95% CI 32.0, 46.2), each FOPL led to an increase in the percentage of participants who correctly identified all products with high levels of nutrient(s) of concern, with the biggest differences observed for the warning label (60.8%, 95% CI 53.5, 68.0; *p* < 0.001), followed by the traffic light label (54.8%, 95% CI 47.9, 61.8; *p* < 0.001), GDA label (55.0%, 95% CI 47.1, 62.9; *p* < 0.001), and HSR label (45.0%, 95% CI 37.2, 52.8; *p* < 0.01). Relative to the control label (2.6, 95% CI 2.5, 2.8), no FOPL led to a statistically significant difference in participants’ intentions to purchase. Full numerical results for both outcomes are shown in Appendix A
Table A6. 

Results on other secondary outcomes can be found in Table 2. Relative to the control group (1.7, 95% CI 1.5, 1.8), each FOPL led to an increase in perceived message effectiveness, with the biggest difference observed for the warning label (2.1, 95% CI 1.9, 2.3; *p* < 0.001), followed by the traffic light label (2.0, 95% CI 1.8, 2.2; *p* < 0.001), GDA label (1.9, 95% CI 1.7, 2.0; *p* < 0.001), and HSR label (1.9, 95% CI 1.8, 2.1; *p* < 0.001).

Relative to the control group, all FOPL types led to increases in perceptions that products were unhealthy, while only the GDA label led to increases in perceptions that products were visually attractive (*p* < 0.05 for all comparisons). With regard to label perceptions, relative to the control group, all FOPLs were rated higher as being attention-grabbing, causing the labels to make participants concerned about the health consequences of consuming the product, be understandable, and teach them something new (*p* < 0.05 for all comparisons). Relative to the control label, all FOPLs except for the HSR were rated higher on being true, while only the GDA and traffic light label were rated higher on being likable. 

### 3.3. Moderation by Sociodemographic and Behavioral Characteristics

Results on moderation of the main outcomes by FOPL type can be found in Table 3 and Table 4. For the ability to correctly identify all products high in nutrient(s) of concern, there was no moderation by any variable except for state (Table 3). For the impact of FOPLs on participants’ ability to correctly identify products high in nutrients of concern, the pattern of results suggested that the impact of FOPLs was greatest in Uttar Pradesh. This state had the biggest differences between the FOPL types (GDA, HSR, MTL) and the control (or second-biggest differences, for warning labels). In contrast, the impact of FOPLs was the smallest in Odisha, with FOPLs either leading to no statistical difference compared with the control (warning labels, GDA, MTL) or a negative difference (HSR). Despite these differences, there was some degree of consistency in the difference between FOPLs and the control across states, with the pattern of results generally showing the biggest differences for warnings, then for GDA or MTL, and relatively small differences for HSR.

For intentions to purchase, there was no moderation by most variables (Table 4). For the HSR label, there was moderation by urbanicity such that the effect of HSR was greater for semi-urban and peri-rural areas than for urban areas (*p* = 0.004).

### 3.4. Label Selection

The results for when participants were asked which label they preferred are shown in Figure 4. Warning labels were most often selected as most likely to discourage consumption of the high-in products by adults, and warnings, GDA, and HSR were most often selected as the most likely to discourage feeding the products to children. Participants selected MTL as the easiest to understand label and GDA as the most informative label.

### 3.5. Sensitivity Analyses

Results from the sensitivity analyses of main outcomes are found in Appendix A
Table A7 and Table A8. In the analyses which removed clustering of standard errors by interviewer, the magnitude and direction of effects remained similar but the estimates became more precise. Relative to the control, warning labels led to a small but statistically significant reduction in participants’ intentions to purchase unhealthy products (p < 0.018). There were no differences in the pattern of results when interviewers with the highest or lowest values were excluded.

## 4. Discussion

This experimental in-person study of adults in six states of India found that, relative to a control label, FOPLs improved the participants’ ability to correctly identify packaged foods and drinks high in nutrients of concern, including sugar, saturated fat, and sodium. The warning label showed the biggest impact on this outcome. Warning labels also showed the biggest impact on several secondary outcomes, including perceived message effectiveness, an outcome that was shown to predict behavioral change [69]. In the main analyses, no FOPL statistically significantly affected intentions to purchase the packaged products, suggesting more research is needed to understand the potential behavior effects of FOPLs. 

The finding that warning labels were the most effective FOPL on the pre-specified primary outcomes in terms of helping consumers identify products high in nutrients of concern was consistent with our prior conceptual framework. This framework for nutrient warnings posits that they are particularly well suited to reducing the consumption of “high-in” products because of their binary nature, which facilitates quick decisions, and their ability to signal a warning, which communicates action (to discourage consumption) [34,70]. 

However, in the main analyses, warning labels failed to statistically significantly reduce intentions to purchase unhealthy packaged products, a finding that was not in line with other empirical evidence in India and elsewhere. For example, a separate recent study conducted on FOPLs among Indian consumers found that, compared with GDA and HSR labels, the warnings led to the biggest impact on intentions to purchase unhealthy products [61]. In addition, a recently published systematic review of randomized controlled trials and quasi-experiments found that warning labels were more effective than color-coded labels (e.g., traffic lights) in discouraging unhealthy food purchasing behavior [35]. 

In contrast, in this study, we found a pattern of results that suggested warning labels had a small effect on reduced purchasing intentions, but these differences did not reach statistical significance after clustering standard errors at the interviewer level. However, these null results should be interpreted with caution given that participants were only exposed to the FOPLs one time in an artificial setting, whereas in a real-world setting, they would be exposed many times. The fact that warnings increased participants’ ability to identify ‘high-in’ products but was not large enough to change behavioral intentions suggests the need to reinforce any FOPL policy with a robust and focused communications campaign to increase consumer awareness and understanding. The lack of statistically significant results may also reflect a lack of precision in our statistical analysis, as warnings statistically significantly reduced intentions to purchase in our sensitivity analysis, when clustering of standard errors was not used (resulting in narrower confidence intervals). Finally, the lack of results could also be due to the use of mock products and lack of realism, which would have affected all arms and potentially blunted any effect of FOPLs on participants’ desire to buy the products. Future research on FOPLs and actual food purchases will be important for understanding their impact on consumer behavior in India. 

It is worth noting that there is some controversy in the field about which public health goals should be prioritized when it comes to the desired outcome of an FOPL system. The current study was designed to test FOPLs’ impacts on antecedents to reducing purchases of foods and drinks high in nutrients of concern because global dietary recommendations consistently agree on the importance of preventing or reducing the consumption of excess amounts of sugar, sodium, saturated fat, and added sugar [71,72,73]. These results suggest that warning labels hold the most promise for helping Indian consumers identify “high-in” products, though these findings should be replicated in a behavioral trial in which actual purchases or intake behaviors are measured.

In contrast, this study found that the HSR system was the lowest-performing FOPL (relative to the control) with regard to helping consumers identify “high-in” products. These findings are consistent with a recent experiment among Colombian adults found that the HSR consistently performed worse than the warning label on multiple outcomes, including identifying “high in” products [74].

One likely reason for the HSR’s low performance is that it was designed with different public goals in mind. The HSR is designed to nudge consumers toward “healthier” purchases, not to identify unhealthy products. Previous studies showed that the HSR helps consumers to rank products based on healthfulness [75] and to help nudge them toward healthier choices [76]. Yet, other studies have shown that for unhealthy products (i.e., those receiving one star or a warning), HSR is less effective than warning labels [77]. 

In addition, it is unclear from a health perspective whether shifting consumers from a lower-scoring product to a higher-scoring product will result in meaningful gains for health. This is of particular concern because in a mandatory HSR system, ultra-processed products could be eligible to carry up to five stars, implying that these products are healthier and should be encouraged, despite a rapidly growing body of evidence from controlled feeding studies and from many prospective cohort studies showing that ultra-processed foods are linked to weight gain, overweight/obesity, and an array of adverse cardiometabolic effects [16,26]. In addition, others have criticized the HSR system for misrepresenting nutrition science [26], in part due to the use of an algorithm that does not reflect how human metabolism works (e.g., the presence of beneficial ingredients such as fiber or protein does not offset the harms of sugar, sodium, or saturated fat). To design an effective FOPL system for Indian consumers, policymakers should consider not only the design of the FOPL itself but also the underlying nutritional profile and health goal (e.g., reducing consumption of the most unhealthy items vs. encouraging healthier (yet likely to be ultra-processed) options).

With regards to secondary outcomes on product perceptions and label reactions, outcomes were mixed, though the pattern suggested a strong performance of warning labels and poor performance for HSR. Warning labels performed best on perceived message effectiveness, a scale that reflects both message perceptions (judgments about how well the message will lead to persuasion) and effects perceptions (how well the message will change behavioral antecedents or the actual behavior) [78,79]. Perceived message effectiveness was used in the development of many health messages across a range of products [80,81,82], and is predictive of behavioral change [69], offering further support for warnings as a strong FOPL to discourage the consumption of “high-in” products among Indian consumers. Warning labels also performed best on other outcomes (identifying products as unhealthy, making participants concerned about health consequences) and similarly to the GDA and/or the MTL on other outcomes (understandable, taught me something new, is true, liking), though the magnitude of difference between the FOPLs was small across all outcomes. The GDA and MTL performed best at grabbing attention. The HSR performed worse than all other FOPL types tested (except the control) on most secondary outcome.

When shown all the FOPLs and asked to select which one they most preferred on a range of outcomes, the results were mixed. Warning labels were most frequently chosen as the label that would most discourage consumption by adults or feeding the products to children. In contrast, the GDA label and MTL label were selected as the most informative and easiest to understand. However, there is no evidence that these preference measures are predictive of behavior change. Conceptually, preference for or “liking” the label may be inversely associated with the intended behavioral change (discouraging purchases), as they may attract consumers toward selecting the product.

With regard to effect modification by socio-demographic and dietary factors, our study found mixed results. Importantly, we found that the impact of FOPLs did not vary for high vs. low educated populations, which suggested that FOPLs hold promise as a population intervention across populations. On the other hand, this study did not assess literacy, and thus, we were not able to understand whether the FOPLs performed well among illiterate populations, which is especially important considering that approximately a quarter of the Indian population is illiterate (and this figure is higher amongst women and in rural areas) [83]. Few studies have looked at FOPLs among illiterate populations. However, principles of visual communication suggest and empirical data shows that imagery can better convey health risk information than can text or numerical information [63,84,85,86], particularly to low literacy groups. This suggests that the warning label (which, in this study, carried icons depicting sugar, salt, and saturated fat) and the HSR (which uses stars) would hold an advantage over labels such as the traffic light or GDA label. Limited empirical data also illustrates the promise of using icons with warnings; in particular, one focus group study in South Africa suggested that warnings with icons would work well among an illiterate population [62]. Other work from the United States found that warnings with icons were perceived as more effective among populations with lower English language literacy [63]. Future research in India should test the effectiveness of FOPLs in populations with low literacy.

We did observe some differences in the impact of FOPL by state. While the pattern of results consistently found that warning labels performed best, the impact of FOPLs was the smallest in the state of Odisha, where FOPLs either had no impact (warnings, GDA, or MTL) or a negative impact (HSR) relative to the control. An effective FOPL policy for India should consider using a state-based educational campaign to ensure that the FOPL is well-understood and used across different populations.

Limitations of this study included that it measured only participants’ self-reported perceptions and reactions. Future experimental trials with more realistic products in more realistic settings that more closely mirror real-world food environments will be necessary. An additional limitation is that we were unable to examine whether FOPLs’ impact varied for people with low incomes or low literacy. Further testing in populations with low incomes and low literacy will be important to ensure that an FOPL system works well for all Indian consumers. The strengths of this study included the large sample and the inclusion of six states (and five languages).

## 5. Conclusions

This randomized field experiment found that, relative to a control label, all FOPLs helped consumers to identify unhealthy packaged products high in sugar, sodium, and saturated fat. The pattern of results suggested that the warning label is the optimal FOPL to achieve the goal of informing consumers about packaged foods and drinks high in nutrients of concern. Replication of this study with behavioral outcomes would provide stronger evidence to support FOPL policies in the Indian population and understand whether such policies would impact consumers’ purchasing behaviors.

## Figures and Tables

**Figure 1 nutrients-14-03128-f001:**
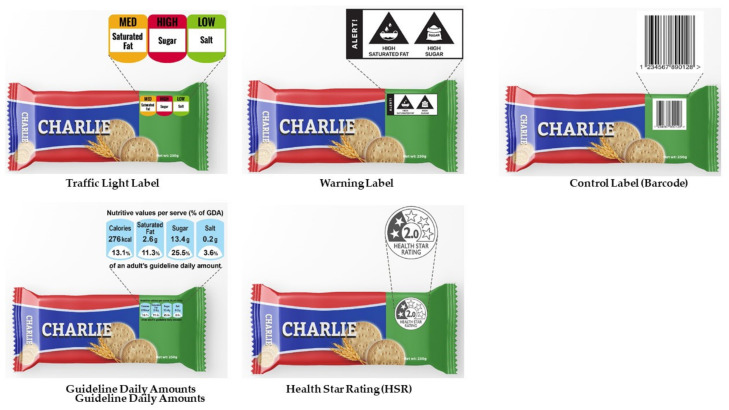
Front-of-package labels (FOPLs).

**Figure 2 nutrients-14-03128-f002:**
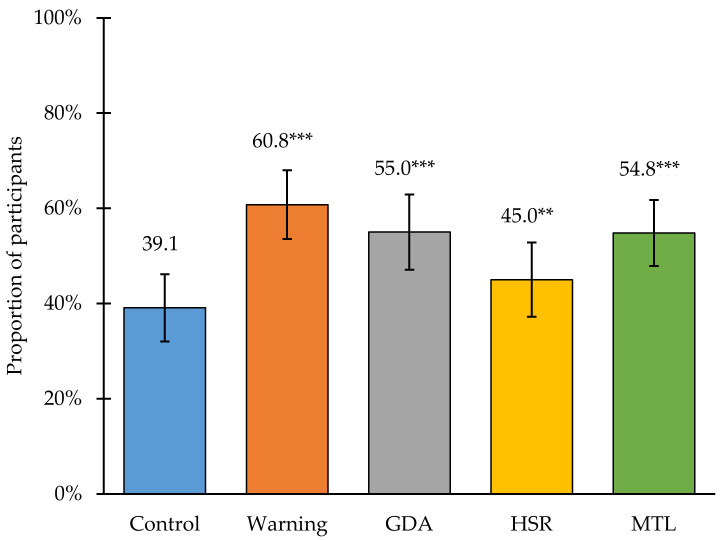
Percent of participants who correctly identified that products were high in nutrient(s) of concern by study arm. *** *p*-Value < 0.001 relative to the control label; ** *p*-Value < 0.01; GDA—Guideline Daily Amount, HSR—Health Star Rating, MTL—Multiple Traffic Light Label.

**Figure 3 nutrients-14-03128-f003:**
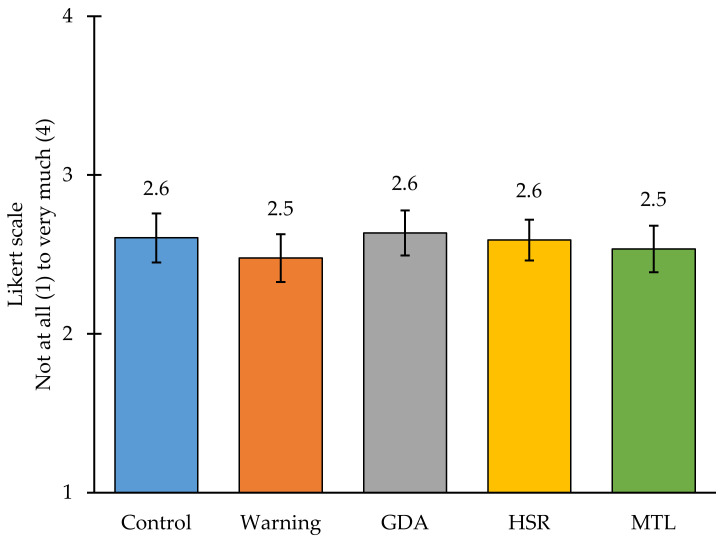
Mean purchase intentions by study arm. GDA—Guideline Daily Amount, HSR—Health Star Rating, MTL—Multiple Traffic Light Label.

**Figure 4 nutrients-14-03128-f004:**
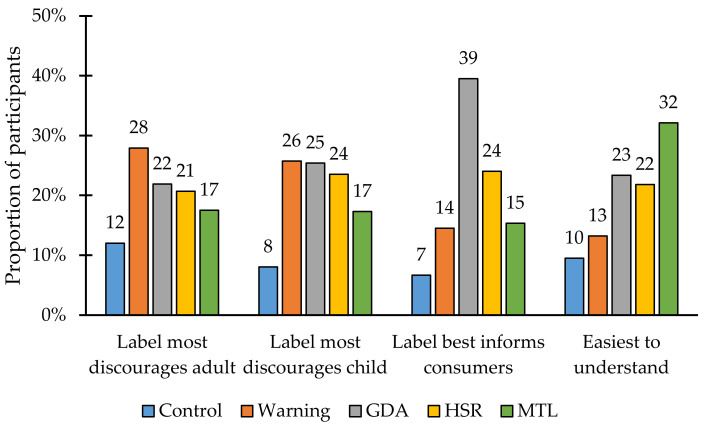
Percent of participants selecting a particular FOPL.

**Table 1 nutrients-14-03128-t001:** Socio-demographic characteristics of the sample, *n* (%).

	*p*	Control (*n* = 574)	Warning (*n* = 598)	GDA (*n* = 554)	HSR (*n* = 601)	MTL (*n* = 542)	Total (*n* = 2869)
State	0.114						
Odisha		79 (13.8)	94 (15.7)	92 (16.6)	88 (14.6)	83 (15.3)	436 (15.2)
Uttar Pradesh		83 (14.5)	89 (14.9)	102 (18.4)	90 (15.0)	91 (16.8)	455 (15.9)
Assam		99 (17.2)	110 (18.4)	92 (16.6)	90 (15.0)	75 (13.8)	466 (16.2)
Delhi		110 (19.2)	82 (13.7)	94 (17.0)	108 (18.0)	97 (17.9)	491 (17.1)
Karnataka		96 (16.7)	119 (19.9)	94 (17.0)	120 (20.0)	84 (15.5)	513 (17.9)
Gujarat		107 (18.6)	104 (17.4)	80 (14.4)	105 (17.5)	112 (20.7)	508 (17.7)
Urbanicity	0.603						
Urban		307 (53.5)	289 (48.3)	286 (51.6)	309 (51.4)	285 (52.6)	1476 (51.4)
Semi-urban		133 (23.2)	168 (28.1)	131 (23.6)	149 (24.8)	124 (22.9)	705 (24.6)
Peri-rural		134 (23.3)	141 (23.6)	137 (24.7)	143 (23.8)	133 (24.5)	688 (24.0)
Age	0.880						
18–30 year		195 (34.0)	209 (34.9)	176 (31.8)	205 (34.1)	190 (35.1)	975 (34.0)
31–40 year		200 (34.8)	220 (36.8)	212 (38.3)	211 (35.1)	187 (34.5)	1030 (35.9)
41–60 year		179 (31.2)	169 (28.3)	166 (30.0)	185 (30.8)	165 (30.4)	864 (30.1)
Gender	0.933						
Man		290 (50.5)	301 (50.3)	286 (51.6)	298 (49.6)	266 (49.1)	1441 (50.2)
Woman		284 (49.5)	297 (49.7)	268 (48.4)	303 (50.4)	276 (50.9)	1428 (49.8)
Education level	0.098						
<12 years		256 (44.6)	237 (39.6)	254 (45.8)	255 (42.4)	254 (46.9)	1256 (43.8)
≥12 years		318 (55.4)	361 (60.4)	300 (54.2)	346 (57.6)	288 (53.1)	1613 (56.2)
Salty biscuit intake	0.279						
<1×/week		171 (29.8)	167 (27.9)	167 (30.1)	188 (31.3)	166 (30.6)	859 (29.9)
1×/week		189 (32.9)	211 (35.3)	159 (28.7)	200 (33.3)	189 (34.9)	948 (33.0)
>1×/week		214 (37.3)	220 (36.8)	228 (41.2)	213 (35.4)	187 (34.5)	1062 (37.0)
Sweet biscuit intake	0.068						
<1×/week		112 (19.5)	112 (18.7)	119 (21.5)	142 (23.6)	118 (21.8)	603 (21.0)
1×/week		164 (28.6)	155 (25.9)	164 (29.6)	158 (26.3)	122 (22.5)	763 (26.6)
>1×/week		298 (51.9)	331 (55.4)	271 (48.9)	301 (50.1)	302 (55.7)	1503 (52.4)
Bread intake	0.696						
<1×/week		137 (23.9)	169 (28.3)	146 (26.4)	156 (26.0)	130 (24.0)	738 (25.7)
1×/week		148 (25.8)	151 (25.3)	145 (26.2)	168 (28.0)	148 (27.3)	760 (26.5)
>1×/week		289 (50.3)	278 (46.5)	263 (47.5)	277 (46.1)	264 (48.7)	1371 (47.8)
Fruit drink intake	0.139						
<1×/week		245 (42.7)	231 (38.6)	248 (44.8)	251 (41.8)	228 (42.1)	1203 (41.9)
1×/week		141 (24.6)	139 (23.2)	139 (25.1)	137 (22.8)	145 (26.8)	701 (24.4)
>1×/week		188 (32.8)	228 (38.1)	167 (30.1)	213 (35.4)	169 (31.2)	965 (33.6)
Noodles intake	0.515						
<1×/week		201 (35.0)	202 (33.8)	206 (37.2)	238 (39.6)	190 (35.1)	1037 (36.1)
1×/week		139 (24.2)	144 (24.1)	139 (25.1)	145 (24.1)	139 (25.6)	706 (24.6)
>1×/week		234 (40.8)	252 (42.1)	209 (37.7)	218 (36.3)	213 (39.3)	1126 (39.2)
Financial situation	0.212						
Excellent		216 (37.6)	247 (41.3)	208 (37.5)	240 (39.9)	200 (36.9)	1111 (38.7)
Good		251 (43.7)	251 (42.0)	259 (46.8)	257 (42.8)	261 (48.2)	1279 (44.6)
Moderate		86 (15.0)	80 (13.4)	57 (10.3)	78 (13.0)	60 (11.1)	361 (12.6)
Poor		21 (3.7)	20 (3.3)	30 (5.4)	26 (4.3)	21 (3.9)	118 (4.1)
Mixed language	0.274						
Yes		231 (40.2)	242 (40.5)	204 (36.8)	251 (41.8)	234 (43.2)	1162 (40.5)

Intake was measured as the self-reported frequency of weekly consumption over the previous 30 days. Financial situation was rated as follows: excellent—can pay the bills, buy the necessary things, and additional things; good—can pay the bills and buy the necessary things; moderate—can pay the bills but not all the necessary things; poor—cannot pay the bills. Mixed language reflects whether the language was conducted bilingually in the native state language and English. The *p*-Values were from chi-square tests for differences by arm.

**Table 2 nutrients-14-03128-t002:** Label reactions and product perceptions by arm, mean (95% CI).

	Control	Warning		GDA		HSR		MTL	
	Mean (95% CI)	Mean (95% CI)	*p*	Mean (95% CI)	*p*	Mean (95% CI)	*p*	Mean (95% CI)	*p*
Product perceptions									
*The product is*…									
Unhealthy	1.7 (1.5, 1.8)	2.1 (1.9, 2.3)	<0.001	1.9 (1.7, 2.2)	0.002	1.8 (1.7, 2.0)	0.003	2.0 (1.8, 2.2)	<0.001
Visually attractive	2.7 (2.5, 2.9)	2.8 (2.7, 3.0)	0.171	2.9 (2.7, 3.1)	0.026	2.8 (2.6, 2.9)	0.233	2.9 (2.7, 3.0)	0.050
Label reactions									
*The label…*									
Grabs my attention	2.7 (2.6, 2.9)	2.9 (2.8, 3.1)	0.031	3.0 (2.8, 3.1)	0.004	2.8 (2.7, 3.0)	0.040	3.0 (2.8, 3.1)	0.004
Makes me concerned about health consequences	1.9 (1.7, 2.1)	2.4 (2.2, 2.6)	<0.001	2.3 (2.0, 2.5)	<0.001	2.2 (2.0, 2.3)	<0.001	2.3 (2.1, 2.5)	<0.001
Is easy to understand	2.4 (2.2, 2.6)	2.8 (2.7, 3.0)	<0.001	2.8 (2.7, 3.0)	<0.001	2.7 (2.6, 2.9)	<0.001	2.8 (2.6, 2.9)	<0.001
Taught me something new	2.3 (2.1, 2.5)	2.8 (2.6, 3.0)	<0.001	2.8 (2.6, 3.0)	<0.001	2.7 (2.5, 2.8)	<0.001	2.7 (2.5, 2.9)	<0.001
Is true	2.6 (2.4, 2.7)	2.9 (2.7, 3.0)	<0.001	2.9 (2.7, 3.1)	<0.001	2.7 (2.5, 2.8)	0.066	2.8 (2.6, 3.0)	0.006
Liking the label	2.7 (2.5, 2.9)	2.9 (2.7, 3.0)	0.072	2.9 (2.8, 3.1)	0.035	2.8 (2.7, 3.0)	0.054	2.9 (2.8, 3.1)	0.006
PME	1.7 (1.5, 1.8)	2.1 (1.9, 2.3)	<0.001	1.9 (1.7, 2.0)	<0.001	1.9 (1.8, 2.1)	<0.001	2.0 (1.8, 2.2)	<0.001

PME—perceived message effectiveness.

**Table 3 nutrients-14-03128-t003:** Effect of FOPL type on the percent of participants who correctly identified products high in nutrients of concern, moderated by socio-demographic characteristics.

	Control	Warning		GDA		HSR		MTL	
	% (95% CI)	% (95% CI)	P^a^	% (95% CI)	P^a^	% (95% CI)	P^a^	Mean (95% CI)	P^a^
Education									
<12 years	35.2 (27.3, 43.1)	51.5 (42.3, 60.6)	<0.001	49.5 (40.3, 58.7)	<0.001	45.2 (36.3, 54.2)	0.002	46.7 (37.9, 55.6)	<0.001
≥12 years	42.3 (34.8, 49.8)	66.6 (59.5, 73.7)	<0.001	59.6 (51.9, 67.3)	<0.001	44.8 (36.3, 53.4)	0.437	61.8 (54.7, 68.9)	<0.001
P^b^		0.073		0.469		0.120		0.074	
Language of interview									
State language	41.5 (34.1, 48.8)	63.3 (55.6, 71.1)	<0.001	54.9 (45.4, 64.4)	0.001	44.6 (35.6, 53.6)	0.279	57.6 (50.8, 64.4)	<0.001
Mixed (state language and English)	35.5 (23.8, 47.3)	57.0 (44.2, 69.7)	0.001	55.2 (42.9, 67.5)	<0.001	45.7 (33.4, 57.9)	0.002	51.0 (38.0, 64.1)	0.001
P^b^		0.955		0.354		0.103		0.910	
Urbanicity									
Urban	40.0 (32.2, 47.7)	59.9 (52.7, 67.1)	<0.001	54.0 (44.4, 63.5)	0.001	47.3 (39.2, 55.3)	0.013	54.6 (47.0, 62.2)	<0.001
Semi-urban	44.1 (36.0, 52.2)	68.0 (59.9, 76.0)	<0.001	58.7 (48.8, 68.7)	0.004	44.8 (34.1, 55.4)	0.858	55.7 (46.2, 65.1)	0.015
Peri-rural	32.2 (24.1, 40.3)	53.8 (44.0, 63.5)	<0.001	53.6 (45.7, 61.5)	<0.001	40.4 (30.4, 50.5)	0.054	54.6 (46.7, 62.4)	<0.001
P^b^		0.678		0.273		0.244		0.135	
Gender									
Men	40.6 (32.3, 48.9)	61.2 (54.4, 68.0)	<0.001	54.2 (45.5, 62.9)	<0.001	46.6 (37.9, 55.2)	0.094	54.3 (46.0, 62.7)	<0.001
Women	37.6 (30.1, 45.2)	60.4 (51.4, 69.3)	<0.001	55.9 (46.4, 65.4)	<0.001	43.5 (34.8, 52.2)	0.054	55.3 (47.5, 63.0)	<0.001
P^b^		0.607		0.316		0.981		0.301	
State									
Odisha	47.1 (32.2, 62.1)	46.4 (30.9, 62.0)	0.923	50.4 (36.7, 64.0)	0.521	39.1 (24.0, 54.2)	0.024	42.7 (25.5, 59.8)	0.322
Uttar Pradesh	54.1 (32.7, 75.5)	86.0 (76.7, 95.2)	0.002	86.0 (77.7, 94.3)	0.001	68.5 (47.1, 90.0)	0.011	78.9 (68.0, 89.8)	0.012
Assam	25.8 (11.3, 40.3)	47.2 (34.7, 59.8)	0.002	33.2 (19.5, 47.0)	0.138	21.9 (6.0, 37.9)	0.121	45.8 (31.2, 60.3)	0.002
Delhi	23.2 (9.1, 37.3)	59.5 (42.5, 76.5)	<0.001	42.6 (27.6, 57.5)	0.037	30.4 (23.1, 37.8)	0.229	42.0 (27.2, 56.8)	0.014
Karnataka	57.4 (43.8, 71.0)	72.2 (58.4, 86.0)	<0.001	60.7 (47.0, 74.4)	0.178	63.4 (51.8, 75.0)	0.026	64.8 (55.4, 74.3)	0.046
Gujarat	33.4 (18.2, 48.6)	53.8 (34.6, 73.1)	0.023	52.4 (34.5, 70.3)	0.063	44.5 (24.2, 64.8)	0.109	53.0 (36.8, 69.2)	0.006
P^b^		0.025		0.027		<0.001		0.002	
Sweet biscuit intake									
<1×/week	25.0 (13.1, 36.9)	51.8 (36.4, 67.2)	<0.001	36.1 (22.7, 49.5)	0.071	27.5 (16.4, 38.6)	0.645	36.4 (27.0, 45.9)	0.050
1×/week	22.0 (14.0, 29.9)	39.4 (27.1, 51.6)	0.002	35.4 (22.1, 48.6)	0.013	27.8 (16.2, 39.5)	0.231	29.5 (20.1, 38.9)	0.095
>1×/week	26.2 (15.9, 36.5)	49.2 (38.7, 59.8)	<0.001	42.4 (29.7, 55.1)	0.001	30.9 (19.6, 42.2)	0.218	36.8 (26.4, 47.2)	0.006
P^b^		0.347		0.732		0.894		0.788	
Bread intake									
<1×/week	42.3 (31.0, 53.7)	66.3 (54.8, 77.7)	<0.001	50.0 (39.4, 60.6)	0.205	50.0 (37.8, 62.2)	0.320	61.5 (52.7, 70.4)	0.002
1×/week	50.0 (37.0, 63.0)	62.9 (48.6, 77.2)	0.091	54.5 (41.9, 67.1)	0.555	50.0 (38.0, 62.0)	1.000	60.8 (48.1, 73.5)	0.110
>1×/week	45.0 (35.2, 54.8)	73.0 (64.8, 81.2)	<0.001	61.6 (50.0, 73.2)	0.001	48.0 (36.7, 59.3)	0.455	62.5 (51.4, 73.6)	0.004
P^b^		0.148		0.292		0.718		0.582	
Fruit drink intake									
<1×/week	58.4 (47.2, 69.6)	75.8 (66.0, 85.5)	<0.001	73.8 (63.0, 84.5)	0.005	69.7 (57.6, 81.8)	0.041	75.0 (62.7, 87.3)	<0.001
1×/week	59.6 (46.6, 72.6)	79.1 (68.7, 89.6)	0.004	77.0 (66.4, 87.6)	0.012	61.3 (44.5, 78.1)	0.798	75.2 (64.4, 86.0)	0.017
>1×/week	63.8 (52.2, 75.5)	76.3 (66.1, 86.5)	0.020	70.7 (58.6, 82.7)	0.264	59.2 (47.0, 71.3)	0.365	70.4 (60.2, 80.6)	0.266
P^b^		0.522		0.363		0.138		0.336	
Noodle intake									
<1×/week	31.3 (20.9, 41.8)	62.9 (52.5, 73.2)	<0.001	48.5 (36.9, 60.2)	0.004	44.1 (30.6, 57.6)	0.028	54.7 (46.5, 63.0)	<0.001
1×/week	29.5 (18.2, 40.8)	54.2 (41.6, 66.8)	<0.001	47.5 (36.7, 58.3)	0.004	44.1 (34.0, 54.3)	0.007	41.7 (30.1, 53.4)	0.070
>1×/week	27.4 (18.4, 36.3)	46.8 (35.0, 58.6)	<0.001	45.9 (33.6, 58.3)	0.002	34.4 (22.9, 45.9)	0.097	53.5 (41.4, 65.6)	<0.001
P^b^		0.207		0.983		0.559		0.218	
Savory biscuit intake									
<1×/week	32.7 (20.8, 44.7)	52.7 (38.9, 66.5)	0.001	52.1 (40.7, 63.4)	<0.001	46.3 (33.9, 58.6)	0.013	53.6 (43.4, 63.9)	0.001
1×/week	40.2 (29.2, 51.2)	59.2 (48.8, 69.7)	0.002	57.2 (45.2, 69.3)	0.010	40.5 (29.6, 51.4)	0.951	51.9 (41.5, 62.2)	0.037
>1×/week	34.6 (22.9, 46.3)	60.0 (47.4, 72.6)	<0.001	61.0 (47.3, 74.7)	<0.001	44.6 (30.3, 58.9)	0.017	57.8 (42.2, 73.3)	0.001
P^b^		0.673		0.344		0.096		0.267	

P^a^ is the *p*-Value for no difference between a given FOPL and the control at a given moderator level. P^b^ is the *p*-Value for equal differences with the control mean across moderation levels within FOPL.

**Table 4 nutrients-14-03128-t004:** Effect of FOPL type on intentions to consume products high in nutrients of concern, moderated by socio-demographic characteristics.

	Control	Warning		GDA		HSR		MTL	
	Mean (95% CI)	Mean (95% CI)	P^a^	Mean (95% CI)	P^a^	Mean (95% CI)	P^a^	Mean (95% CI)	P^a^
Education									
<12 years	2.6 (2.4, 2.8)	2.5 (2.3, 2.7)	0.408	2.6 (2.5, 2.8)	0.507	2.6 (2.5, 2.8)	0.743	2.5 (2.4, 2.7)	0.405
≥12 years	2.6 (2.5, 2.8)	2.4 (2.3, 2.6)	0.014	2.6 (2.5, 2.8)	0.875	2.6 (2.4, 2.7)	0.433	2.5 (2.4, 2.7)	0.256
P^b^		0.298		0.647		0.503		0.941	
Language of interview									
State language	2.7 (2.5, 2.9)	2.6 (2.4, 2.8)	0.086	2.7 (2.5, 2.9)	0.801	2.7 (2.5, 2.8)	0.446	2.6 (2.4, 2.8)	0.159
Mixed (state language and English)	2.4 (2.2, 2.7)	2.3 (2.1, 2.5)	0.115	2.5 (2.3, 2.7)	0.284	2.5 (2.3, 2.6)	0.606	2.4 (2.2, 2.6)	0.906
P^b^		0.720		0.329		0.377		0.312	
Urbanicity									
Urban	2.6 (2.4, 2.8)	2.4 (2.3, 2.6)	0.005	2.6 (2.5, 2.8)	0.793	2.5 (2.3, 2.6)	0.033	2.5 (2.4, 2.7)	0.256
Semi-urban	2.6 (2.4, 2.7)	2.6 (2.4, 2.8)	0.643	2.7 (2.6, 2.9)	0.053	2.7 (2.5, 2.8)	0.139	2.5 (2.3, 2.6)	0.299
Peri-rural	2.6 (2.4, 2.9)	2.4 (2.2, 2.6)	0.033	2.6 (2.4, 2.8)	0.508	2.7 (2.6, 2.9)	0.312	2.6 (2.4, 2.9)	0.944
P^b^		0.055		0.208		0.004		0.780	
Gender									
Men	2.6 (2.4, 2.8)	2.5 (2.3, 2.7)	0.113	2.7 (2.5, 2.9)	0.705	2.6 (2.4, 2.8)	0.741	2.6 (2.4, 2.8)	0.501
Women	2.6 (2.4, 2.8)	2.4 (2.3, 2.6)	0.070	2.6 (2.5, 2.8)	0.728	2.6 (2.4, 2.7)	0.992	2.5 (2.3, 2.7)	0.235
P^b^		0.765		0.916		0.806		0.788	
State									
Odisha	2.4 (2.1, 2.7)	2.2 (2.0, 2.5)	0.123	2.2 (1.9, 2.4)	0.061	2.2 (2.0, 2.5)	0.033	2.1 (1.8, 2.4)	0.002
Uttar Pradesh	2.7 (2.2, 3.1)	2.6 (2.3, 3.0)	0.945	2.8 (2.4, 3.1)	0.610	2.7 (2.4, 3.0)	0.976	2.8 (2.4, 3.1)	0.552
Assam	3.1 (2.9, 3.3)	2.8 (2.7, 3.0)	0.043	3.1 (2.9, 3.3)	0.951	3.1 (2.8, 3.3)	0.714	3.0 (2.9, 3.1)	0.217
Delhi	2.3 (2.1, 2.6)	2.0 (1.6, 2.3)	0.004	2.5 (2.2, 2.8)	0.250	2.4 (2.1, 2.6)	0.663	2.3 (2.1, 2.5)	0.910
Karnataka	2.7 (2.5, 2.9)	2.8 (2.5, 3.1)	0.362	2.8 (2.6, 3.0)	0.322	2.7 (2.5, 2.9)	0.707	2.7 (2.4, 3.1)	0.847
Gujarat	2.4 (2.0, 2.9)	2.2 (1.8, 2.6)	0.015	2.4 (2.1, 2.7)	0.900	2.5 (2.1, 2.8)	0.808	2.4 (2.1, 2.7)	0.608
P^b^		0.056		0.304		0.492		0.177	
Sweet biscuit intake									
<1×/week	2.6 (2.3, 2.8)	2.5 (2.2, 2.7)	0.555	2.5 (2.3, 2.8)	0.845	2.5 (2.3, 2.7)	0.654	2.3 (2.1, 2.5)	0.048
1×/week	2.6 (2.4, 2.8)	2.5 (2.3, 2.7)	0.341	2.7 (2.5, 2.9)	0.470	2.7 (2.5, 2.8)	0.337	2.7 (2.5, 2.9)	0.407
>1×/week	2.7 (2.5, 2.9)	2.5 (2.3, 2.7)	0.056	2.7 (2.5, 3.0)	0.675	2.7 (2.5, 2.9)	0.884	2.6 (2.4, 2.9)	0.500
P^b^		0.820		0.718		0.627		0.070	
Bread intake									
<1×/week	2.5 (2.3, 2.8)	2.4 (2.1, 2.7)	0.470	2.5 (2.3, 2.7)	0.768	2.5 (2.2, 2.7)	0.515	2.2 (2.0, 2.4)	0.029
1×/week	2.6 (2.3, 2.9)	2.4 (2.1, 2.7)	0.079	2.5 (2.3, 2.6)	0.207	2.6 (2.4, 2.7)	0.716	2.6 (2.3, 2.8)	0.779
>1×/week	2.7 (2.5, 2.9)	2.6 (2.4, 2.8)	0.527	2.8 (2.6, 3.0)	0.212	2.8 (2.7, 3.0)	0.112	2.7 (2.4, 2.9)	0.663
P^b^		0.724		0.142		0.325		0.140	
Fruit drink intake									
<1×/week	2.5 (2.3, 2.8)	2.4 (2.1, 2.7)	0.271	2.6 (2.4, 2.8)	0.437	2.5 (2.3, 2.7)	0.996	2.3 (2.1, 2.6)	0.100
1×/week	2.5 (2.3, 2.8)	2.4 (2.2, 2.6)	0.485	2.5 (2.3, 2.7)	0.876	2.6 (2.4, 2.8)	0.470	2.6 (2.4, 2.8)	0.491
>1×/week	2.7 (2.5, 2.8)	2.5 (2.3, 2.7)	0.094	2.7 (2.5, 3.0)	0.453	2.7 (2.5, 2.8)	0.974	2.6 (2.4, 2.9)	0.940
P^b^		0.765		0.886		0.790		0.220	
Noodle intake									
<1×/week	2.5 (2.3, 2.7)	2.4 (2.1, 2.7)	0.556	2.4 (2.2, 2.7)	0.826	2.4 (2.2, 2.6)	0.337	2.3 (2.1, 2.5)	0.033
1×/week	2.6 (2.3, 2.8)	2.4 (2.2, 2.6)	0.096	2.7 (2.5, 2.9)	0.159	2.5 (2.3, 2.7)	0.559	2.6 (2.4, 2.8)	0.950
>1×/week	2.6 (2.3, 2.9)	2.5 (2.3, 2.6)	0.203	2.6 (2.5, 2.8)	0.746	2.6 (2.3, 2.8)	0.882	2.6 (2.4, 2.8)	0.818
P^b^		0.782		0.500		0.840		0.208	
Savory biscuit intake									
<1×/week	2.5 (2.3, 2.7)	2.5 (2.3, 2.7)	0.886	2.5 (2.3, 2.8)	0.606	2.4 (2.2, 2.7)	0.639	2.5 (2.3, 2.7)	0.710
1×/week	2.7 (2.4, 2.9)	2.5 (2.4, 2.7)	0.263	2.6 (2.4, 2.8)	0.592	2.6 (2.4, 2.8)	0.449	2.7 (2.5, 2.9)	0.959
>1×/week	2.8 (2.5, 3.0)	2.5 (2.3, 2.8)	0.012	2.7 (2.5, 3.0)	0.957	2.8 (2.6, 3.0)	0.834	2.6 (2.3, 3.0)	0.325
P^b^		0.329		0.739		0.783		0.636	

P^a^ is the *p*-Value for no difference between a given FOPL and the control at a given moderator level. P^b^ is the *p*-Value for equal differences with the control mean across moderation levels within FOPL.

## Data Availability

De-identified data can be found at Open Science Framework at https://osf.io/.

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
