# Peer review of "Front-of-Package Labels on Unhealthy Packaged Foods in India: Evidence from a Randomized Field Experiment"

_nutrients, 2022, doi:10.3390/nu14153128_

Round 1

Reviewer 1 Report

Dear Authors,

Thank you for the opportunity to review this research dealing with front-of-package nutrition labels among Indian consumers. The issue represents a challenge for the EU ongoing debate with reference to a mandatory Front-of- Package labelling model.

I have some concerns which I have to address you before I can suggest the publication of the article to Nutrients.

The title is too broad: please change it so to better highlights your interesting research in less words.

The literature you cite is not so updated (manuscript published in 2022 are a few) and, please, avoid to cite yourself so many times. Do not cite manuscripts under review. We still do not know if they will be accepted.

Please edit the manuscript carefully following the “Instructions for Authors”. For example, in the text, reference numbers should be placed in square brackets [ ], and placed before the punctuation and please indicate before the definition and after the acronyms (Line 117).

Line 119-120: why BRANY (Biomedical Research Alliance of New York) is written in bold?

Line 122: This study was pre-registered at Open Science Framework in December 2022. December 2022? How is it possible?

Conclusion have to be rewritten and improved.

Reviewer 2 Report

This paper compared different front-of-pack labels, a topical issue with ongoing discussion on the most appropriate label to use. I consider this paper to be clear and comprehensive using appropriate methodology. The discussion considered the use of the different labels, such as acknowledging the different purpose of Health Star Ratings.

There are some minor errors in the text (see below), and the presentation of most of the tables could be improved. 

92: add year

93: add correct citation

117: Replace IRB with a heading without an acronym

138: the list

145: participants

145-147: Please review this sentence, what is TLC, what is the number 28 referring to?

181: healthy

Table 2 and 3: Have the CIs on a separate line to the mean. It is difficult to scan the results and compare. Table 4 results are presented clearly. Left justify the row headings in the first column. 

520: Nutritional

546: may be inversely

Discussion: You comment on needing to assess if the FOPLs would perform well with people who are illiterate. Do you think the labels would perform well with people who can not pay bills, given this was a very small proportion of your sample?
